# Contextual factors predicting compliance behavior during the COVID-19 pandemic: A machine learning analysis on survey data from 16 countries

Nandor Hajdu[1,2]*, Kathleen Schmidt[3], Gergely Acs[4], Jan P. Röer[5], Alberto Mirisola[6], Isabella Giammusso[6], Patrícia Arriaga[7], Rafael Ribeiro[7], Dmitrii Dubrov[8], Dmitry Grigoryev[8], Nwadiogo C. Arinze[9], Martin Voracek[10], Stefan Stieger[11], Matus Adamkovic[12,13], Mahmoud Elsherif[14], Bettina M. J. Kern[10,15], Krystian Barzykowski[16], Ewa Ilczuk[16], Marcel Martončik[12], Ivan Ropovik[17,18], Susana Ruiz-Fernandez[19,20], Gabriel Baník[12], José Luis Ulloa[21], Balazs Aczel[2‡], Barnabas Szaszi[2‡]

1 Doctoral School of Psychology, ELTE, Eötvös Loránd University, Budapest, Hungary, 2 Institute of Psychology, ELTE, Eötvös Loránd University, Budapest, Hungary, 3 Ashland University, Ashland, Ohio, United States of America, 4 Department of Networked Systems and Services, Budapest University of Technology and Economics, Budapest, Hungary, 5 Department of Psychology and Psychotherapy, Witten/ Herdecke University, Witten, Germany, 6 Department of Psychology, Educational Science and Human Movement, University of Palermo, Palermo, Italy, 7 ISCTE-University Institute of Lisbon, CIS-IUL, Lisboa, Portugal, 8 National Research University Higher School of Economics, Moscow, Russian Federation, 9 Alex Ekwueme Federal University, Ndufu-Alike, Abakaliki, Nigeria, 10 Department of Cognition, Emotion, and Methods in Psychology, Faculty of Psychology, University of Vienna, Vienna, Austria, 11 Division Psychological Methodology, Department of Psychology and Psychodynamics, Karl Landsteiner University of Health Sciences, Krems an der Donau, Austria, 12 Institute of Psychology, Faculty of Arts, University of Presov, Prešov, Slovakia, 13 Institute of Social Sciences, CSPS Slovak Academy of Sciences, Košice, Slovakia, 14 Department of Psychology, University of Birmingham, Birmingham, United Kingdom, 15 Department of European and Comparative Literature and Language Studies, Faculty of Philological and Cultural Studies, University of Vienna, Vienna, Austria, 16 Institute of Psychology, Faculty of Philosophy, Jagiellonian University, Krakow, Poland, 17 Faculty of Education, Charles University, Prague, Czech Republic, 18 Faculty of Education, University of Presov, Prešov, Slovakia, 19 FOM University of Applied Sciences, Essen, Germany, 20 Leibniz-Institut für Wissensmedien, Tübingen, Germany, 21 Programa de Investigación Asociativa (PIA) en Ciencias Cognitivas, Centro de Investigación en Ciencias Cognitivas (CICC), Facultad de Psicología, Universidad de Talca, Talca, Chile

‡ BA and BS are joint senior authors on this work.
* hajdu.nandor93@gmail.com

**Data Availability Statement:** The data and analysis script are available at https://osf.io/dfsvb/.

## Abstract

Voluntary isolation is one of the most effective methods for individuals to help prevent the transmission of diseases such as COVID-19. Understanding why people leave their homes when advised not to do so and identifying what contextual factors predict this non-compliant behavior is essential for policymakers and public health officials. To provide insight on these factors, we collected data from 42,169 individuals across 16 countries. Participants responded to items inquiring about their socio-cultural environment, such as the adherence of fellow citizens, as well as their mental states, such as their level of loneliness and bore-dom. We trained random forest models to predict whether someone had left their home during a one week period during which they were asked to voluntarily isolate themselves. The analyses indicated that overall, an increase in the feeling of being caged leads to an

**Funding:** Balazs Aczel, Nandor Hajdu and Barnabas Szaszi were supported by the Hungarian National Research, Development and Innovation Office (NKFIH-1157-8/2019-DT); Gabriel Baník was supported by APVV-17-0418; Patrícia Arriaga was supported by the Portuguese National Funding Agency for Science and Technology (FCT, REF UID/PSI/03125/2020).; Ivan Ropovik was supported by PRIMUS/20/HUM/009; Matus Adamkovic was supported by the Slovak Research and Development Agency [project no. APVV-20-0319]; Dmitry Grigoryev and Dmitrii Dubrov were supported by the HSE University Basic Research Program; Krystian Barzykowski and Ewa Ilczuk were supported by the National Science Centre, Poland (UMO-2019/35/B/HS6/00528). The research reported in this paper is part of project no. BME-NVA-02, implemented with the support provided by the Ministry of Innovation and Technology of Hungary from the National Research, Development and Innovation Fund, financed under the TKP2021 funding scheme. The funders had no role in study design, data collection and analysis, decision to publish, or preparation of the manuscript.

**Competing interests:** Author Martin Voracek is a PLOS ONE Editorial Board Member. This does not alter the author's adherence to PLOS ONE Editorial policies and criteria.

increased probability of leaving home. In addition, an increased feeling of responsibility and an increased fear of getting infected decreased the probability of leaving home. The models predicted compliance behavior with between 54% and 91% accuracy within each country's sample. In addition, we modeled factors leading to risky behavior in the pandemic context. We observed an increased probability of visiting risky places as both the anticipated number of people and the importance of the activity increased. Conversely, the probability of visiting risky places increased as the perceived putative effectiveness of social distancing *decreased*. The variance explained in our models predicting risk ranged from < .01 to .54 by country. Together, our findings can inform behavioral interventions to increase adherence to lockdown recommendations in pandemic conditions.

## Introduction

When no treatment or vaccine is available to prevent transmission, behavioral measures may be the most effective means of containing a disease [1], though some researchers consider it not as beneficial [2]. One such approach is to ensure that people minimize contact with other individuals, either by keeping a safe distance from other people in public places or by staying at home. To increase this behavior, governments can initialize lockdowns, and set rules that regulate for what purpose people can meet. However, maintaining sufficient compliance with these rules and regulations is difficult, especially for extended periods of time [3]. In order to counter the spread of a disease, understanding which factors influence people's compliance with confinement recommendations is essential.

Among all the different factors that could affect staying at home during a pandemic (e.g. personality traits), contextual factors are the focus of the present paper. We define contextual factors as the physical and sociocultural environment along with the intrapersonal circumstances, such as mental states, present at the time of the choice that may affect decisions. Contextual factors can accurately predict decisions in simple situations where most of this contextual information can be identified [4]. Identifying the contextual factors of non-adherence to lockdown recommendations and exploring their relative predictive strength will provide insight into decisions that put individuals and communities at risk. These insights can help public health officials and policy makers design interventions to target the factors that have the largest effect on decision making. Although not labeled as such in previous research, many factors that fit our definition of contextual factors (e.g. confidence in the government to tackle the pandemic [5]) have already been studied. However, the literature is limited regarding the systematic investigation of the contextual factors that influence people's decisions to comply with confinement regulations.

Most lockdown regulations during the Covid-19 pandemic have allowed individuals to leave their residences for essential reasons. The definitions of what constitutes an essential or non-essential activity varies according to region, but most regulations or recommendations classify going to work (when working from home is not possible), attending school or another educational institution, shopping for groceries and medicine, seeking medical care, and exercise as essential activities that justify venturing outside (e.g., [6]). Outings for any other reason are considered to be non-essential activities (e.g. social gatherings). Here, we consider leaving home for non-essential reasons as non-compliant behavior during lockdown.

## Mental states and beliefs as context

Some of the main factors that motivate individuals to leave their home during confinement are feelings of loneliness [7] and other unpleasant mental states. Boredom is also a prevalent state during social isolation and boredom proneness is a critical risk factor for non-compliance with social-distancing protocols [8]. Further, adverse reactions to recommendations or requirements to stay inside may lead to feelings of captivity. This sentiment is well reflected in the oft-used metaphor of "being imprisoned" when people describe their situation during quarantine [9]. These mental states likely decrease adherence to social isolation recommendations during lockdown.

General compliance with isolation rules or recommendations also appears to be influenced by attitudes and beliefs, such as thinking that taking health precautions is effective against the infection [10]. Among these beliefs, perceived vulnerability, beliefs that getting COVID-19 would be disruptive, and government trust each have very small positive effects on general compliance [10]. However, other factors such as trust in policies seem to have stronger effects. Researchers have found increased mobility reduction—thus, compliance with quarantine regulations—in European regions where the levels of trust in policymakers prior to the COVID-19 pandemic was high [11]. In a study exploring the effects of self-perceived risk of contracting COVID-19, fear of the virus, moral foundations, and political orientation on compliance with public health recommendations, only fear emerged as a predictor of compliance [12]. The perceived infectiousness of COVID-19 may also have an effect on rule compliance; the more contagious people think COVID-19 is, the less willing they are to take social distancing measures. This counterintuitive relationship has been described as the "fatalism effect" [13]. Finally, the sense of duty and responsibility could also contribute to staying at home [14] because leaving the house would be perceived as irresponsible.

Motivation to remain at home during requested social isolation periods can stem from trusting in someone or something. People might not leave their homes because they trust the regulations to be effective or place their trust in a higher power [15]. Also, generalized social trust appears to moderate the indirect effect of personality traits on rule-respecting behaviors; individuals who trust others demonstrate more compliance than those who do not [16]. Expert opinion may also motivate compliance; providing people with expert information about the spreading of the virus partially corrects their misconceptions about transmission [13]. Compliant people seem to perceive protective measures as effective, while non-compliant people perceive them as problematic [17]. Altogether, several factors have emerged as potential predictors of non-compliant behavior in the context of the current pandemic. However, these factors have not been examined systematically across cultures.

The present research was designed to extend the literature on lockdown regulations by systematically investigating the contextual factors that influence compliance in confinement situations across cultures. First, we conducted a pilot study to identify potential contextual factors that might affect compliance with confinement recommendations. Then, in our main study, we explored how these factors influenced the behavior of participants from 16 countries using a machine learning approach. Specifically, we tested the extent to which these factors predict *(a)* compliance with confinement recommendations and *(b)* the risk-taking behaviors of non-compliant individuals.

## Pilot study

The main goal of the pilot study was to identify potential influencing factors that might have an effect on whether or not someone stays at home during a pandemic. A brief survey was used to collect qualitative data to achieve this goal.

## Methods

The research plan was approved by the lead authors' local institutional ethical review board, the Research Ethics Committee of ELTE PPK. The survey respondents were recruited from a university participant pool in Hungary that consisted of students of various undergraduate and graduate programs who received course credit as compensation. Participants gave their informed, written consent to take part in the study. The survey was conducted in March 2020, three weeks after the lockdown measures were first locally imposed. Participants responded to open-ended questions about what influences their decisions and those of other people when they choose to leave their home and go to a place where they might be in close physical proximity to others. To process the answers, we used inductive coding to compare responses to factors already derived from the existing literature or generated by brainstorming. For each answer, the first author decided whether the given answer contained a new type of factor. If a newly processed answer could not be labeled as belonging to any of the registered categories, a new category was created.

## Results

A total of 532 participants completed the survey. After processing all the responses, we added 1 additional factor that may influence adherence to confinement recommendations, for a total of 25: *being afraid of getting infected; feeling that staying home is the responsible behavior; feeling caged; being afraid of the consequences of getting infected; being afraid of infecting someone else; thinking that they are already a vector; feeling lonely; feeling bored; thinking that the pandemic will have serious economic consequences; belief in the effectiveness of social distancing; being in contact with elderly/someone with chronic illness; country leaders' communication; trust in a higher power; trust in experts' opinion; trust in people who attend the out-of-home activity; knowing people who attend the out-of-home activity; event importance; peers' opinion; family opinion; number of people attending the out-of-home activity; possibly meeting many people while getting to the site of the out-of-home activity; out-of-home activity site size; event is indoors or outdoors*; and *level of hygiene at the location of the out-of-home activity*. The additional item was *being up-to-date about the virus*. The collected factors represent the opinions and thoughts of Hungarian university students. While their answers are not representative of the world, it seems plausible that the potential influencing factors they indicated have an effect in other countries, too.

## Main study

The goal of our main study was to explore the extent to which the factors identified in the pilot study predict compliance with lockdown recommendations. Also, we investigated whether the riskiness of an out-of-home activity can be predicted from contextual factors, such as the spaciousness of the place or other circumstances.

## Methods

The methods and analyses for the main study were pre-registered and can be found at https://osf.io/7nfu8. Deviations from the pre-registration are detailed in the S1 File. The research plan was approved by the lead authors' local institutional ethical review board, the Research Ethics Committee of ELTE PPK. The data were collected between April 29, 2020 and November 12, 2020.

## Participants

Participants were recruited with the collaboration of 16 research labs, and gave their informed, written consent to take part in the study. Each research lab organized individual campaigns of participant recruitment through various media outlets, university participant pools, or paid participant pools. Details of recruitment methods for each lab can be found in the S1 File. In total, we recruited 43,123 participants from 102 countries; however, we only analyzed data from the 16 countries with more than 100 respondents ($n$ = 42,169) to allow for more complex and more robust analyses. The countries included in the study were: Austria, Germany, Greece, Hungary, Italy, Japan, the Netherlands, Nigeria, Poland, Portugal, Romania, Russia, Slovakia, Switzerland, the UK, and the USA.

## Materials and procedures

The study was conducted online via Qualtrics. First, respondents reported their age, gender, years of education, country of residence, monthly income, and the number of people in their household. Then, participants were asked if they had left their home in the previous 7 days of the lockdown for non-essential reasons. There was a slight difference in wording between countries where there was a lockdown at the time of response and where the lockdown had already ended. In cases where there was a lockdown at the time of response, the question was: *"Did you leave your home in the last 7 days for non-essential reasons?."* Where the country did not have any restrictions in effect at the time of the survey, the question was the following: *"Did you leave your home in the last 7 days of the lockdown for non-essential reasons? Lockdown is the period when residents in your region were asked not to leave their homes for non-essential reasons."* Participants were informed that essential reasons included: buying groceries or medicine, going to work, and seeking medical attention in case of serious illness/injury. Next, participants were asked to indicate the degree to which the statements—corresponding to each of the 24 factors identified in the pilot study—applied to them or to their activity on a 7-point Likert-type scale (1 = *did not apply at all*; 7 = *completely applied*).

Event-specific items that referred to factors concerning the context of the out-of-home activity only appeared for participants who actually left their home during the investigated period. For these event-specific items, participants were asked to respond to statements about their most recent non-essential out-of-home activity. The 9 event-specific items measured were: *peer pressure to take part in the activity; the number of people present; degree of acquaintance; trust in the people present; preconception about how many people they would meet; location size; location indoors or outdoors; hygiene of the location*; and *importance of the activity*.

Event-general items (i.e., those not specific to an out-of-home activity) were shown to every respondent, regardless of whether they left their homes in the previous 7 days. For these items, participants were asked to indicate their degree of agreement with 16 statements describing *the fear of getting infected; thought that already contacted the virus, boredom, loneliness, coping with being indoors, thoughts about symptom seriousness if infected, economic consequences, putative effectiveness of social distancing, trust in a higher power, contact with elderly or someone with chronic illness, fear of infecting someone else, feeling of responsibility, encouragement of country leaders, encouragement of experts, adherence of fellow citizens, being up-to-date about the virus.* Note that the "*contact with elderly or someone with chronic illness*" and the "*being up-to-date about the virus*" items were excluded from analyses by the lead team because they were judged not to measure context. Among these items, participants also responded to an attention-check item: *"I went to the Moon twice."*

The original English language questionnaire was translated to eleven languages by native speakers from the participating research labs. The full survey for each language is available at https://osf.io/u38zh/.

## Data analysis

To answer the question of why people leave their homes during a pandemic lockdown, we opted to use random forest models, a machine learning method [18]. Random forests are popular prediction algorithms for several reasons: they are robust to the non-linearity of data, they do not require data to be normalized, and they typically provide superior prediction accuracy while mitigating overfitting without extensive parameter tuning. It is a standard method of machine learning and is frequently employed when the number of variables to consider is relatively low. However, this method has some limitations. The results are not as easy to interpret because decision trees are stochastic, which means that they can change with different runs. Random forests are made of decision trees. Each decision tree in the forest is a set of internal nodes and leaves. In the internal node, a feature is selected along which the data is split into two groups. Then, each group is subdivided iteratively, following the same rule until some condition is met on the size of the tree or the number of data points in the node. For classification problems, the criterion to select a feature can be Gini impurity or information gain. We used information gain in our calculations. The average information gain increase is collected for each feature selected for the splits. The average of this increase over all trees in the forest is the measure of variable importance. Because a random subset of features is used for a tree, the result is also random. However, if we have many trees, then the resulting importance values should be similar to one another. We analyzed data from each country separately.

To explore the factors that predict non-compliance (i.e., leaving home for non-essential reasons), we created random forest models using the event-general items and demographic variables such as age, gender, income, and years of education. Data were split into training and test sets in an 80–20 ratio. On our training dataset, the number of variables in each division of a tree node was between 2 and 10 and were tuned separately for every country via 10-fold cross-validation. Then, we tested how well each model performed on the test data by calculating classification accuracies. We also calculated variance importance metrics for each model. These metrics inform us of the degree of importance of a variable to predict outcomes. We used the variance importance scores based on the mean decrease in accuracy when the given variable is removed from the model.

To analyze the riskiness of activities, we first defined a "risk" score as the sum in the levels of crowdedness, size, level of hygiene, and whether the event was indoors. The greater this score the higher the risk of the activity. Next, we created random forest regression models on data from individuals who indicated that they left their homes during the lockdowns. Consequently, we could include both event-specific and event-general items in this analysis. We use the risk score as the dependent variable to estimate the influence of a factor in the decision to participate in an activity despite it being risky. Variable importance was calculated the same way as in the case of non-compliance prediction. The greater the importance of the predictor, the more influence it has on the decisions of people to go outside despite being in a risky situation. As the dependent variable was continuous, we calculated the Root Mean Squared Errors to assess the model accuracy, and chose the model with the lowest error during hyperparameter tuning.

## Results

Data of respondents who did not finish the questionnaire were excluded from the analysis ($N$ = 13,653), along with those who failed the attention check ($N$ = 2,387). We also excluded those who reported the top 0.1% income in each country ($N$ = 43), because the values were unrealistically high. Then, we excluded people who did not identify themselves as either female or male ($N$ = 114 across all countries), because we would not have been able to give reliable predictions on a country level about non-binary people based on this small sample. As a last step in our exclusion procedure, we omitted the data of those countries from where we received no more than 100 responses. The final sample used in the analysis included 42,169 people from 16 countries ($M_{age}$ = 40.91 years, $SD_{age}$ = 12.06, 50.99% female). Table 1 shows the basic descriptive information for each analyzed country.

### Factors predicting non-compliance

A heatmap showing the differences in relative importance for each item and country is shown in Fig 1. As shown, *the fear of getting infected* was in the top three most important factors in 12 out of 16 countries, suggesting that it is one of the most important factors overall in predicting home confinement. The *feeling of responsibility*, the *feeling of being caged while at home*, and *perceived countrymen adherence* also had a great impact on staying at home, as they were in the top three most important factors in 11, 8 and 8 countries, respectively.

We calculated the permutation importance of a variable, i.e., the decrease in prediction accuracy when the given variable is randomized, while other variables are left intact. This randomization was conducted 100 times, and the average importance is reported. To provide a visual representation of the differences between the importance values of variables, we rescaled the variable importance values per country to values between 0 (least important) and 100 (most important). The shade of the color is based on the rescaled importance score, grouped by country: the higher the permutation importance score of a variable in a given country, the darker the color.

**Table 1. Sample descriptive statistics by country.** Left home proportion represents the proportion of people who left their homes for non-essential reasons out of all respondents.

| Country | N | Female proportion | Left home proportion | Median income per month (USD) | Median age (years) | Median years of education |
|---|---|---|---|---|---|---|
| Austria | 1129 | 0.68 | 0.42 | 2739.00 | 28 | 17 |
| Germany | 2215 | 0.67 | 0.47 | 3834.60 | 27 | 16 |
| Greece | 135 | 0.75 | 0.59 | 1314.72 | 50 | 16 |
| Hungary | 35012 | 0.49 | 0.52 | 1987.34 | 42 | 17 |
| Italy | 473 | 0.71 | 0.19 | 657.36 | 28 | 17 |
| Japan | 278 | 0.45 | 0.29 | 1885.92 | 45.5 | 16 |
| Netherlands | 117 | 0.56 | 0.64 | 4930.20 | 35 | 17 |
| Nigeria | 185 | 0.52 | 0.43 | 92.87 | 26 | 14 |
| Poland | 376 | 0.70 | 0.46 | 482.32 | 23 | 16 |
| Portugal | 381 | 0.65 | 0.46 | 2191.20 | 33 | 16 |
| Romania | 115 | 0.50 | 0.38 | 1591.52 | 41 | 17 |
| Russian Federation | 376 | 0.39 | 0.40 | 649.00 | 30 | 15 |
| Slovakia | 349 | 0.86 | 0.36 | 1643.40 | 21 | 15 |
| Switzerland | 150 | 0.53 | 0.62 | 10358.40 | 40 | 18 |
| United Kingdom | 457 | 0.49 | 0.39 | 4449.78 | 38 | 17 |
| USA | 421 | 0.47 | 0.51 | 9500.00 | 36 | 16 |

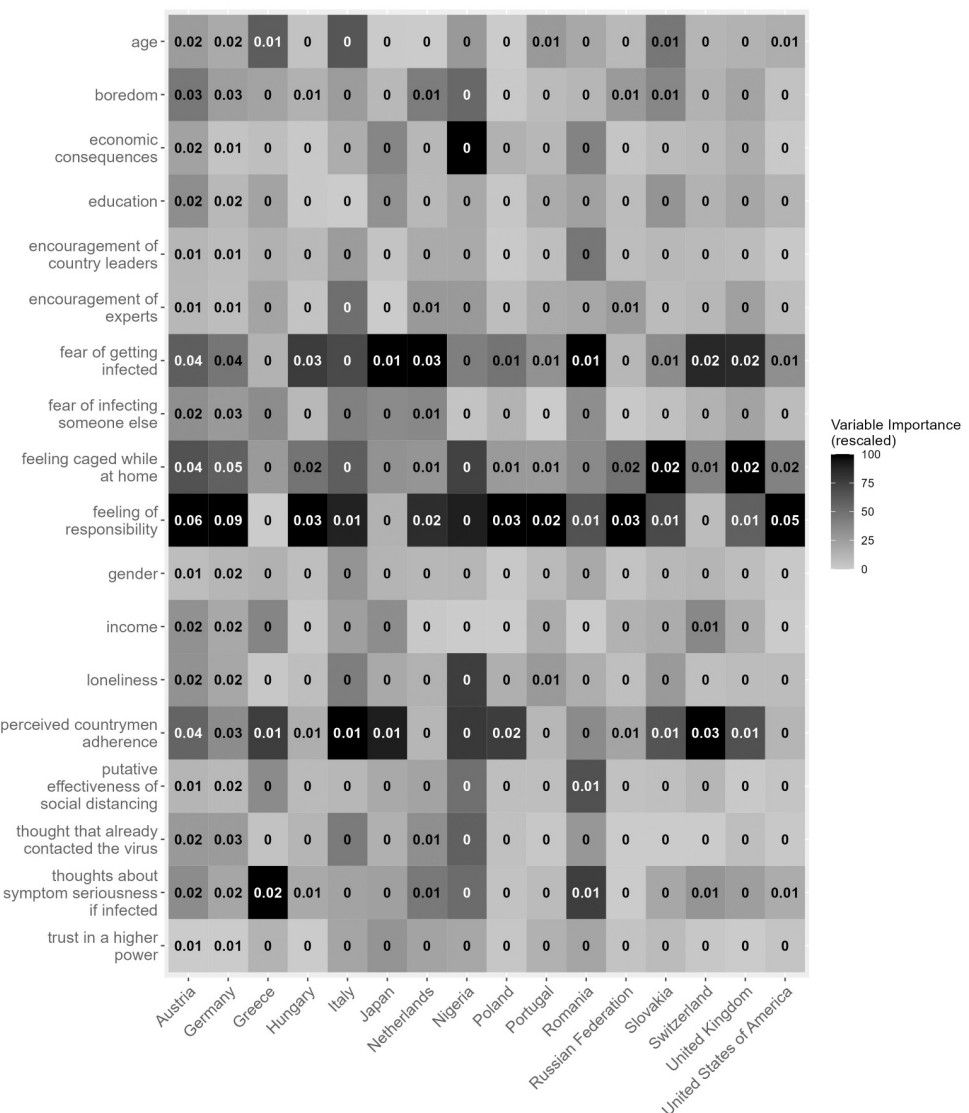

**Fig 1. Variable importance values when predicting leaving home in each country.**

Our models, based on event-general factors, were successful in predicting whether someone left their home during lockdown. Predictions were the most accurate on Austrian data, where 91% of the test cases were classified correctly. The least accurate predictions were made on Nigerian data, with only a 54% accuracy. All the model accuracies are reported in Table 2.

We created partial dependence plots to examine whether a factor was associated with an increased or decreased probability of leaving home (Fig 2). The plots suggest that the general patterns of the results were similar between countries. Inspecting the plots of the top 3 most important variables revealed that scores on the *Feeling of responsibility* scale are negatively related to the probability of non-adherence; *Fear of getting infected* seems to decrease the probability of leaving one's home, while *Feeling caged while at home* increases the probability of leaving one's home.

We calculated how the overall prediction changes at different values of a variable by substituting real data with the same value for every participant and then calculating the mean of

**Table 2. Prediction accuracies of random forest models by country.** *Left home—accuracy* represents the percentage of correct classifications on the test set when predicting whether someone left their home. *Risk—Root Mean Squared Error* indicates the accuracy of predictions on the test set when predicting riskiness of the activity when someone left their home, while *risk—$R^2$* represents the proportion of variance explained by the model.

| Country | left home—accuracy | risk—Root Mean Squared Error | risk—$R^2$ |
|---|---|---|---|
| Austria | 0.91 | 0.72 | 0.52 |
| Germany | 0.84 | 0.67 | 0.54 |
| Greece | 0.55 | 1.06 | 0.11 |
| Hungary | 0.71 | 0.90 | 0.20 |
| Italy | 0.84 | 1.00 | 0.08 |
| Japan | 0.71 | 1.15 | 0.03 |
| Netherlands | 0.75 | 0.92 | 0.15 |
| Nigeria | 0.54 | 0.90 | 0.02 |
| Poland | 0.64 | 0.90 | 0.07 |
| Portugal | 0.64 | 1.09 | 0.10 |
| Romania | 0.65 | 0.74 | <0.01 |
| Russian Federation | 0.71 | 0.90 | 0.21 |
| Slovakia | 0.71 | 0.80 | 0.32 |
| Switzerland | 0.80 | 0.99 | 0.309 |
| United Kingdom | 0.74 | 1.05 | 0.08 |
| United States | 0.66 | 0.94 | 0.14 |

these predictions. This method is appropriate because the variables are uncorrelated. As a result, these predictions for different plugged-in values can be represented on a graph to see how the predictions change from one value of the independent variable to the next. Lines on Fig 2 show the average predicted probability of leaving home associated with a given value of the contextual factor in each country.

## Factors predicting participation in risky activities

After analyzing the factors involved in leaving home during the lockdown, we set out to investigate the factors associated with participation in risky activities. We report the root mean squared errors and $R^2$ values of the final models in Table 2. Variance importance metrics were calculated for each model. A heatmap of variable importance among countries is presented in Fig 3. The results suggest that the *anticipated number of people met while traveling, putative effectiveness of social distancing, activity importance* and *trust in the people met at the activity* are the most important factors when predicting the participation in risky activities. These variables are in the top three most important variables in 8, 6, 5 and 5 countries, respectively.

Similar to Figs 1 and 4 shows the partial dependence plots displaying the level of riskiness associated with each factor and the change in the predicted risk score when a given variable was altered, for each country separately. The plots suggest that the general pattern of the results was similar among countries, and that, in most cases, a change in any one variable amounted to very little change in predicted risk.

## Discussion

The research presented here explored the importance of contextual factors in predicting decisions to stay at home during pandemic lockdowns. The factors we measured appeared to either increase or decrease the probability of leaving home across samples. In fact, the observed variables showed a consistent pattern of prediction across the 16 investigated countries, suggesting

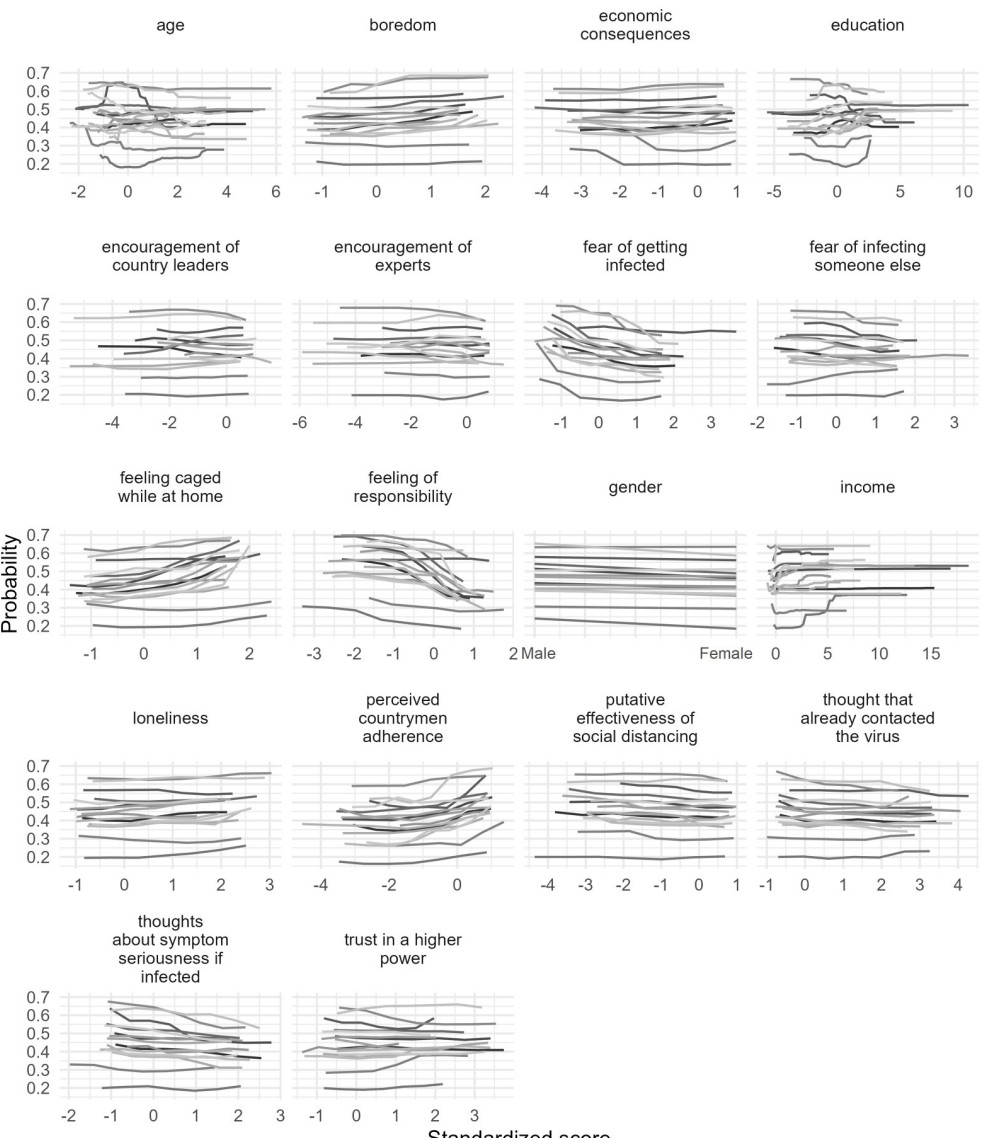

**Fig 2. Partial dependence plots of variables used in the prediction of leaving home for all countries.** Each line represents a different country.

that our findings are robust in developed and developing countries. *Boredom* and the *adherence of fellow citizens to regulations* increased the probability of leaving home in every country, while the *fear of getting infected* and the *feeling of responsibility* decreased the probability of leaving home in every country.

Although the examined countries differed in which factors were most important in predicting compliance with stay at home orders, some factors emerged as highly important in most of our samples. The *fear of getting infected* was among the top 3 most important factors in 12 countries, but its predictive effects on leaving home were particularly accentuated in Hungary, Japan, the Netherlands, Romania, Switzerland, and the UK and comparatively minimal in Greece, Nigeria, and the Russian Federation. *Feeling of responsibility* was one of the top three

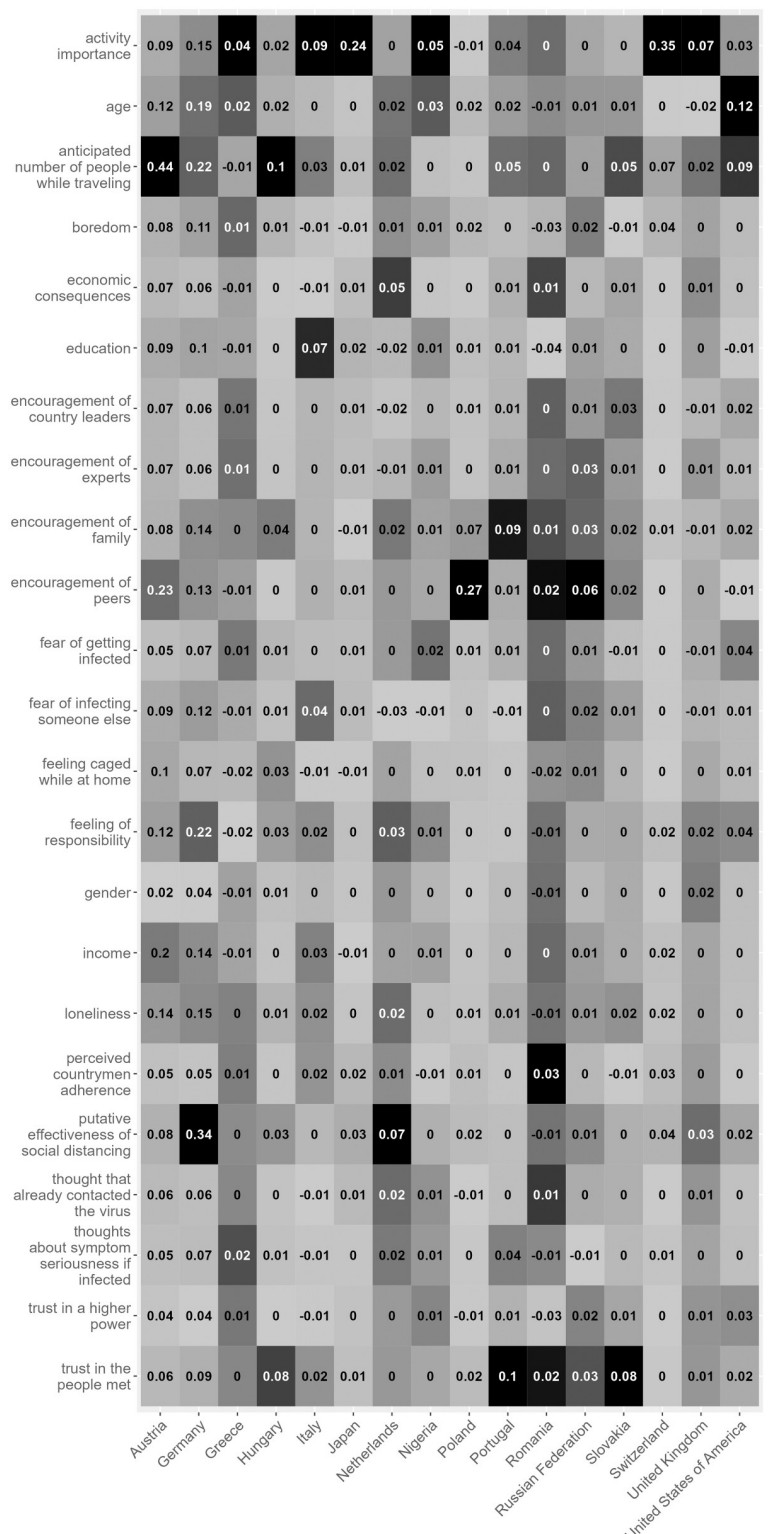

**Fig 3. Variable importances when predicting risk level of out-of-home activity in each country.** The color of each cell is based on variable importance rescaled to the 0–100 range, while numbers in cells represent the original variable importance.

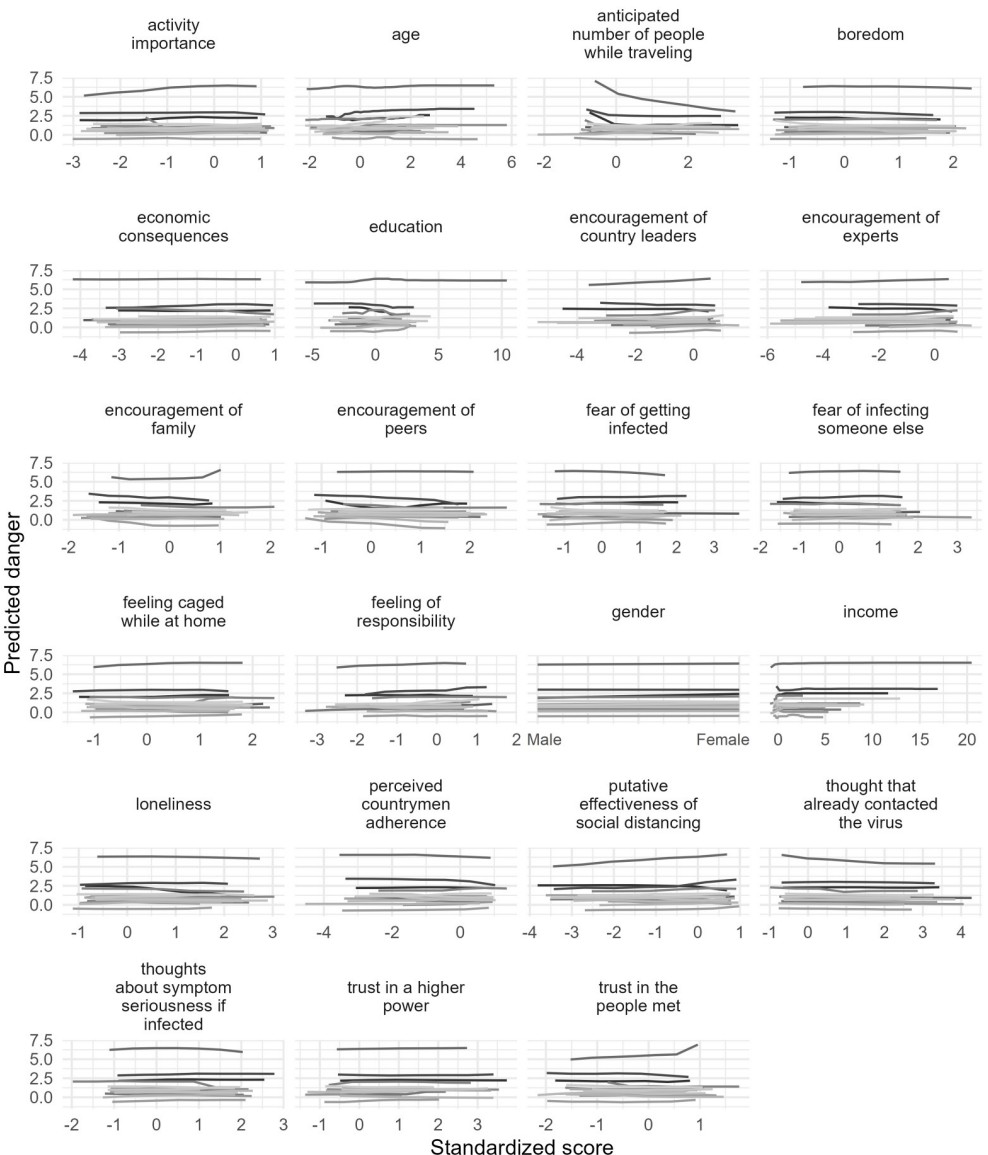

**Fig 4. Partial dependence plots of variables used in the prediction of risk scores for all countries.**

most important factors for 11 countries. This finding suggests that feelings of obligation toward society in preventing the spread of disease increased adherence to confinement recommendations. Relative to other factors, responsibility seemed to have the second largest overall predictive importance: when people feel responsible, they tend to stay home. However, *responsibility* had a strikingly small relationship with adherence in Greece, Japan, and Switzerland. While the *responsibility* factor was not important for these countries, the *perceived countrymen adherence* factor mattered to a great degree. Importantly, prediction accuracy was quite low for these three countries compared to the other countries overall. Perhaps other factors not explored in our study better explain compliance in these nations. Although the *feeling of being caged while at home* was among the top three most important factors in 8 countries, its effects were particularly important in the UK and Slovakia and unimportant in Japan and Greece.

Previous research has demonstrated that mental states, such as the feeling of loneliness [7] and boredom [8] are predictors of non-compliant behavior. Fear of the virus [12] has also been linked to increased compliance, along with the feeling of responsibility [14]. Our study confirmed these effects and showed they are similar across countries.

Our analyses of the factors predicting activity riskiness for those who left their homes showed quite different accuracies between countries. The variables that appeared most frequently in the top 3 most important factors by country were the *anticipated number of people met while traveling* (i.e., 8 out of 16 countries), *activity importance* (i.e., 6 countries), *trust in people met* (i.e., 5 countries), and *putative effectiveness of social distancing* (i.e., 5 countries). The increases in the *anticipated number of people met while traveling* were associated with increased activity risk. The *anticipated number of people met while traveling* was the most important factor in Austria, Hungary, and the second most important in Slovakia and in the USA. This factor was particularly not important in Poland and Russia, however. *Activity importance* was a strong predictor of activity risk in most countries. Seemingly, as activity importance increased, the riskiness of the activity decreased. *Trust in people met* also had a negative relationship with the riskiness of the activity. These latter two findings suggest that individuals who leave their homes for non-essential but important activities with people they trust may be minimizing their risk-taking. Based on the root mean squared error values and $R^2$ values in Table 2, our models were not always accurate in predicting the risk (i.e., risk of infection) of out-of-home activities. In countries with large sample sizes, the models were generally more accurate and accounted for more of the overall variance than in countries with relatively small sample sizes. Compared to predicting when people left their homes, however, the importance of the measured factors in predicting activity riskiness varied more widely.

While the present study explored 14 potential predictors of non-adherence to lockdown recommendations, our research is limited by the exclusion of unidentified contributors. National development levels, cultural differences, and ethnic differences are important measures that might have an effect on compliance, but are not accounted for in our models. Although we took age, years of education, income and gender into consideration, there may be other demographic factors, such as marital status and occupation that could help in creating more nuanced models. Further, additional context-specific factors that contribute to the riskiness of an out-of-home activity may have yielded stronger or more consistent predictions than the factors we included. Our operationalization of activity risk likewise limits our conclusions. The context and sample differences between countries are also worth noting. The sample sizes, data collection methods, rates of infection, and lockdown recommendations varied between (and sometimes within) countries. The inaccurate risk score predictions might be a consequence of relatively low sample sizes in some of the countries. Also, not all countries were in a lockdown during data collection, which means that in some cases we had to rely on how the participants remembered their situation. Furthermore, our sample of countries is limited to where the authors could conduct data collection. Third-world countries are underrepresented in our sample, where countermeasures against COVID-19 are very different to what is possible in developed and developing countries [19]. Stemming from the recruitment methods used, our sample might not be representative, and this leads to our findings being less generalizable.

Overall, we can conclude that the *fear of getting infected* is the most important predictor of adherence to lockdown recommendations, along with *feelings of responsibility* about the transmission of a disease, *feeling caged at home*, and the *perceived adherence of countrymen*. These results have important public health implications. Messaging to convince people to stay home during lockdown should appeal to personal responsibility. Perhaps, compliance could be increased with an intervention stressing that every person has an active role in a pandemic situation and that staying at home is a valuable and important contribution. Attempts to decrease

social isolation and reframe confinement in a positive light (e.g., as a chance for introspection) may also prove effective. A transparent and thorough coverage of symptoms, infection rates, and the possible risks that arise when contracting the disease may also help people reevaluate their priorities and motivate them to comply with confinement regulations.

## Supporting information

**S1 File. It contains all the supporting tables and figures.**
(DOCX)

## Author Contributions

**Conceptualization:** Nandor Hajdu, Jan P. Röer, Balazs Aczel, Barnabas Szaszi.

**Data curation:** Nandor Hajdu, Dmitrii Dubrov.

**Formal analysis:** Nandor Hajdu.

**Funding acquisition:** Patrícia Arriaga.

**Investigation:** Nandor Hajdu, Kathleen Schmidt, Jan P. Röer, Alberto Mirisola, Isabella Giammusso, Patrícia Arriaga, Rafael Ribeiro, Dmitry Grigoryev, Nwadiogo C. Arinze, Martin Voracek, Stefan Stieger, Matus Adamkovic, Mahmoud Elsherif, Bettina M. J. Kern, Krystian Barzykowski, Ewa Ilczuk, Marcel Martončik, Ivan Ropovik, Susana Ruiz-Fernandez, Gabriel Baník, Barnabas Szaszi.

**Methodology:** Nandor Hajdu, Gergely Acs, Balazs Aczel, Barnabas Szaszi.

**Project administration:** Nandor Hajdu, Krystian Barzykowski.

**Resources:** Nandor Hajdu, Kathleen Schmidt, Alberto Mirisola, Isabella Giammusso, Dmitry Grigoryev, Martin Voracek, Stefan Stieger, Matus Adamkovic, Mahmoud Elsherif, Bettina M. J. Kern, Krystian Barzykowski, Marcel Martončik, Ivan Ropovik, Susana Ruiz-Fernandez, Gabriel Baník, Barnabas Szaszi.

**Software:** Nandor Hajdu, Gergely Acs.

**Supervision:** Nandor Hajdu, Krystian Barzykowski.

**Validation:** Nandor Hajdu, Dmitrii Dubrov, José Luis Ulloa, Balazs Aczel, Barnabas Szaszi.

**Visualization:** Nandor Hajdu.

**Writing – original draft:** Nandor Hajdu.

**Writing – review & editing:** Nandor Hajdu, Kathleen Schmidt, Gergely Acs, Jan P. Röer, Alberto Mirisola, Isabella Giammusso, Patrícia Arriaga, Rafael Ribeiro, Dmitry Grigoryev, Nwadiogo C. Arinze, Martin Voracek, Stefan Stieger, Matus Adamkovic, Mahmoud Elsherif, Bettina M. J. Kern, Krystian Barzykowski, Ewa Ilczuk, Marcel Martončik, Ivan Ropovik, Susana Ruiz-Fernandez, Gabriel Baník, José Luis Ulloa, Balazs Aczel, Barnabas Szaszi.

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
