## [Decision Letter · Decision Letter 0]

3 Dec 2021

PONE-D-21-27819Contextual factors predicting compliance behavior during the COVID-19 pandemic: A machine learning analysis on survey data from 16 countriesPLOS ONE

Dear Dr. Hajdu,

Thank you for submitting your manuscript to PLOS ONE. After careful consideration, we feel that it has merit but does not fully meet PLOS ONE’s publication criteria as it currently stands. Therefore, we invite you to submit a revised version of the manuscript that addresses the points raised during the review process.

We look forward to receiving your revised manuscript.

Kind regards,

Donrich Thaldar

Academic Editor

PLOS ONE

Journal Requirements:

Reviewers' comments:

Reviewer's Responses to Questions

**Comments to the Author**

1. Is the manuscript technically sound, and do the data support the conclusions?

Reviewer #1: Partly

Reviewer #2: Yes

2. Has the statistical analysis been performed appropriately and rigorously? 

Reviewer #1: Yes

Reviewer #2: Yes

3. Have the authors made all data underlying the findings in their manuscript fully available?

Reviewer #1: Yes

Reviewer #2: Yes

4. Is the manuscript presented in an intelligible fashion and written in standard English?

Reviewer #1: Yes

Reviewer #2: Yes

5. Review Comments to the Author

Reviewer #1: This paper investigates contextual factors for COVID-19 stay at home orders. The study includes a large sample size and is well conducted. The conclusions are, however, not concrete, details below. I believe the paper can be redirected with some revision.

1) The beginning of the paper refers to isolation, but later on lockdown is referred. It is not clear whether the study refers to isolation due to exposure to COVID or to government instructions, or both?

2) ln 68: What does [1], but see [2] mean?

3) lns 90 - 97: This is not the same across the world - this should be addressed.

4) ln 126: Fix the language

5) Page 6 introduces the novelty of the paper as examining the factors across cultures. However, the data collected is not representative of cultures internationally, nor are cultures extensively analysed with the data. The novelty of the work is thus not strong enough. Publication without the novelty is difficult to justify.

6) The pilot study is very specific to Hungary. This should be addressed. The degrees the students were registered for are also not mentioned in the pilot study. This can create bias and must be addressed. No ethics approval for the pilot study is mentioned.

7) ln 161: Why was an additional response added and which is it?

8) The countries in the study have almost no third world representativeness (Nigeria is present but is one of the strongest countries in Africa economically). The response to COVID has been vastly different is these parts of the world, as a literature search will quickly reveal. The study does not address this anywhere and states in the discussion the results can be extended to all cultures. This is a very false statement without any evidence in the data or analysis. The choice of countries in the study is not discussed. The study should redirect to the data, not an international generalisation (as stated in ln 369). lns 376 - 377 especially are not applicable to those who do not have the means to lockdown. The conclusions in lns 428-438 are shortsighted to the international variance in COVID-19 responses and experiences.

9) lns 224-230: No third world issues are addressed here. The study is thus directed only to people who have the means to lockdown.

10) ln 270: indoors

11) ln 284: Which values were drastically high?

12) The data has 35097 respondents from Hungary. The study is thus heavily biased to Hungary.

13) The data has information on income. The analysis does not seem to take this into account though?

14) ln 308: There is surely a better word for 'darkness'?

15) Covariates could be considered in the analysis to explain differences between countries.

16) Check the paper for consistent spelling: colour vs. color etc.

17)

Reviewer #2: The manuscript trained random forest models to predict which contextual factors predict non-compliance based on data from 42,283 individuals in 16 countries. The article found that increased feelings of being caged led to an increased probability of leaving home, while increased responsibility and increased fear of being infected reduced the probability of leaving home. Further, an increase in the expected number of people and importance of the activity, as well as a decrease in the perceived effect of social distance, increased the probability of visiting risky places.

Overall, this manuscript is very interesting. Weighing the contributions and limitations of this manuscript, I recommend that this manuscript could be accepted for publication in PLOS ONE should the authors be prepared to incorporate some revisions.

1. Make sure there are no spelling or grammatical errors, for example, in line 58 of the Abstract section, "county" should be "country".

2. I would like the authors to present a table (which can be combined with S1 Table 1) showing the lockdown status of the sample countries during the data collection period.

3. Variable importance measures depend on the set of included variables, so omitted variables can affect the ranking of variables in terms of importance. National development levels, cultural differences, and ethnic differences are all factors that should be considered when conducting cross-country comparative studies. Also, demographic factors (e.g., age, education level, occupation, marital status, income level) can affect non-compliance. Finally, the effectiveness of isolation measures is related to their uptake. I hope that the authors provide more detail on unidentified contributors in the limitations section.

4. Recruited respondents may not be matched to their national population in terms of main attributes (e.g., gender, age, residential location), so we cannot deny the possibility that their inclusion in the study may have led to selection bias. This point should be raised in the limitations section.

6. PLOS authors have the option to publish the peer review history of their article (what does this mean?). If published, this will include your full peer review and any attached files.

Reviewer #1: No

Reviewer #2: No

---

## [Author Response · Author response to Decision Letter 0]

13 Apr 2022

Dear Editor and Reviewers,

We are pleased to submit a revision of our manuscript for publication in PLOS One.

We would like to thank you and the reviewers for their constructive comments and helpful suggestions. Below, you can find a point-by-point response to all comments in bold. We look forward to your comments.

Kind regards,

Nandor Hajdu, on behalf of the co-authors

Reviewer #1: This paper investigates contextual factors for COVID-19 stay at home orders. The study includes a large sample size and is well conducted. The conclusions are, however, not concrete, details below. I believe the paper can be redirected with some revision.

1) The beginning of the paper refers to isolation, but later on lockdown is referred. It is not clear whether the study refers to isolation due to exposure to COVID or to government instructions, or both?

Thank you for pointing this out. We added a sentence to the first paragraph that connects isolation to lockdowns: “To increase this behavior, governments can initialize lockdowns, and set rules that regulate for what purpose people can meet.” (page 4).

2) ln 68: What does [1], but see [2] mean?

We reformulated this sentence to better incorporate the second citation into the text.

3) lns 90 - 97: This is not the same across the world - this should be addressed.

In order to accentuate that the definition of non-essential activities is not the same across the world, we changed “The definitions of what constitutes an essential or non-essential activity likely varies according to region …” to “The definitions of what constitutes an essential or non-essential activity varies according to region” (page 5).

4) ln 126: Fix the language

The sentence has been corrected and now reads as: “Motivation to remain at home during requested social isolation periods can stem from trusting in someone or something.” (page 6).

5) Page 6 introduces the novelty of the paper as examining the factors across cultures. However, the data collected is not representative of cultures internationally, nor are cultures extensively analysed with the data. The novelty of the work is thus not strong enough. Publication without the novelty is difficult to justify.

Thank you for your remark. We agree that the data are not representative of cultures, thus we changed the sentence in line 134 to the following: “However, these factors have not been examined systematically and concurrently in one study, across a large number of countries.” We hope that this sentence clarifies that our emphasis was put on the simultaneous exploration of these factors. This is the greatest novelty. In this manner, the effects on “staying at home” are controlled intrinsically. Our methodology, regarding the collection of the potential influencing factors and applying machine learning techniques in this context is also an asset.

6) The pilot study is very specific to Hungary. This should be addressed. The degrees the students were registered for are also not mentioned in the pilot study. This can create bias and must be addressed. No ethics approval for the pilot study is mentioned.

Thank you for pointing this out. We added information about the education of students: “The survey respondents were recruited from a university participant pool in Hungary that consisted of students of various undergraduate and graduate programs who received course credit as compensation.” To indicate that the sample is not representative and might be biased, we added the following: “The collected factors represent the opinions and thoughts of Hungarian university students. While their answers are not representative of the world, it seems plausible that the potential influencing factors they indicated have an effect in other countries, too.”

We also added a statement about the ethics approval for the pilot study.

7) ln 161: Why was an additional response added and which is it?

To clarify this, we added the following to the manuscript: “The additional item was being up-to-date about the virus.” The reason we added this item is that at the time of analyzing the pilot data, we thought that the degree of how well-informed people are regarding the virus could be an important factor in predicting their behavior. However, we ultimately decided against using this predictor in our models, as it did not fit our definition of a contextual factor.

8) The countries in the study have almost no third world representativeness (Nigeria is present but is one of the strongest countries in Africa economically). The response to COVID has been vastly different in these parts of the world, as a literature search will quickly reveal. The study does not address this anywhere and states in the discussion the results can be extended to all cultures. This is a very false statement without any evidence in the data or analysis. The choice of countries in the study is not discussed. The study should redirect to the data, not an international generalisation (as stated in ln 369). lns 376 - 377 especially are not applicable to those who do not have the means to lockdown. The conclusions in lns 428-438 are shortsighted to the international variance in COVID-19 responses and experiences.

Thank you for pointing out this error. We changed the sentence in line 369 from “... suggesting that our findings are robust and may be generalizable across cultures.” to “... suggesting that our findings are robust in developed and developing countries.” We also added this to the limitations paragraph starting at line 416: “Furthermore, our sample of countries is limited to where the authors could conduct data collection. Third-world countries are underrepresented in our sample, where countermeasures against COVID-19 are very different to what is possible in developed and developing countries (Fosu & Edunyah, 2020). Thus, our study can inform about the behavior of people who have the means to lockdown only. “

9) lns 224-230: No third world issues are addressed here. The study is thus directed only to people who have the means to lockdown.

Thank you for making us aware of this shortcoming of our study. This issue is brought up in our previous comment.

10) ln 270: indoors

11) ln 284: Which values were drastically high?

The lowest excluded value was 1 752 960 USD, while the highest was 3.057 * 10126 USD. There was a mistake in the reported number of exclusions based on income that has been corrected.

12) The data has 35097 respondents from Hungary. The study is thus heavily biased to Hungary.

Our analyses would be biased if we pooled all of our data and fitted only one model to explain adherence to confinement. That is why we created separate models for every country. This way, the sample size in one country didnot influence the results in another country. A large sample size from Hungary only means that the model fitted on Hungarian data is the most robust, but it does not weigh more in our comparative analyses.

13) The data has information on income. The analysis does not seem to take this into account though?

We decided to re-run the analysis and included income, along with gender, years of education and age, as well.

14) ln 308: There is surely a better word for 'darkness'?

Thank you for pointing this out. We changed the word ‘darkness’ to ‘shade of color’ .

15) Covariates could be considered in the analysis to explain differences between countries.

Thank you for this suggestion. We re-run the analyses and included four covariates that we had available: age, years of education, gender, and income. These changes are incorporated into the manuscript.

16) Check the paper for consistent spelling: colour vs. color etc.

Thank you for pointing out these inconsistencies. We checked the manuscript for consistent spelling, and made changes where required.

17)

Reviewer #2: The manuscript trained random forest models to predict which contextual factors predict non-compliance based on data from 42,283 individuals in 16 countries. The article found that increased feelings of being caged led to an increased probability of leaving home, while increased responsibility and increased fear of being infected reduced the probability of leaving home. Further, an increase in the expected number of people and importance of the activity, as well as a decrease in the perceived effect of social distance, increased the probability of visiting risky places.

Overall, this manuscript is very interesting. Weighing the contributions and limitations of this manuscript, I recommend that this manuscript could be accepted for publication in PLOS ONE should the authors be prepared to incorporate some revisions.

1. Make sure there are no spelling or grammatical errors, for example, in line 58 of the Abstract section, "county" should be "country".

Thank you for pointing out these inconsistencies. We checked the manuscript for consistent spelling, and made changes where required.

2. I would like the authors to present a table (which can be combined with S1 Table 1) showing the lockdown status of the sample countries during the data collection period.

3. Variable importance measures depend on the set of included variables, so omitted variables can affect the ranking of variables in terms of importance. National development levels, cultural differences, and ethnic differences are all factors that should be considered when conducting cross-country comparative studies. Also, demographic factors (e.g., age, education level, occupation, marital status, income level) can affect non-compliance. Finally, the effectiveness of isolation measures is related to their uptake. I hope that the authors provide more detail on unidentified contributors in the limitations section.

Thank you for this suggestion. We re-run the analyses and included four covariates that we had available: age, years of education, gender, and income. These changes are reflected in the manuscript. Unfortunately, we did not collect data on other demographic factors that might have had an effect, such as occupation and marital status. We acknowledged this limitation in the discussion: “National development levels, cultural differences, and ethnic differences are important measures that might have an effect on compliance, but are not accounted for in our models. Although we took age, years of education, income and gender into consideration, there may be other demographic factors, such as marital status and occupation that could help in creating more nuanced models. “

4. Recruited respondents may not be matched to their national population in terms of main attributes (e.g., gender, age, residential location), so we cannot deny the possibility that their inclusion in the study may have led to selection bias. This point should be raised in the limitations section.

Thank you for pointing this out. As you suggested, we raised this point in the limitations section: “Stemming from the recruitment methods used, our sample might not be representative, and this leads to our findings being less generalizable.”

---

## [Decision Letter · Decision Letter 1]

16 May 2022

PONE-D-21-27819R1Contextual factors predicting compliance behavior during the COVID-19 pandemic: A machine learning analysis on survey data from 16 countriesPLOS ONE

Dear Dr. Hajdu,

Thank you for submitting your manuscript to PLOS ONE. After careful consideration, we feel that it has merit but does not fully meet PLOS ONE’s publication criteria as it currently stands. Therefore, we invite you to submit a revised version of the manuscript that addresses the points raised during the review process.

Could you please improve the graphics quality? This is the only outstanding issue. 

We look forward to receiving your revised manuscript.

Kind regards,

Donrich Thaldar

Academic Editor

PLOS ONE

Journal Requirements:

Reviewers' comments:

Reviewer's Responses to Questions

**Comments to the Author**

1. If the authors have adequately addressed your comments raised in a previous round of review and you feel that this manuscript is now acceptable for publication, you may indicate that here to bypass the “Comments to the Author” section, enter your conflict of interest statement in the “Confidential to Editor” section, and submit your "Accept" recommendation.

Reviewer #1: All comments have been addressed

2. Is the manuscript technically sound, and do the data support the conclusions?

Reviewer #1: Yes

3. Has the statistical analysis been performed appropriately and rigorously? 

Reviewer #1: Yes

4. Have the authors made all data underlying the findings in their manuscript fully available?

Reviewer #1: Yes

5. Is the manuscript presented in an intelligible fashion and written in standard English?

Reviewer #1: Yes

6. Review Comments to the Author

Reviewer #1: Thank you for the revision.

My only additional comment is that the graphics quality need to be improved.

7. PLOS authors have the option to publish the peer review history of their article (what does this mean?). If published, this will include your full peer review and any attached files.

Reviewer #1: No

---

## [Author Response · Author response to Decision Letter 1]

13 Sep 2022

The resolution of Figures 1-4 has been increased.

---

## [Decision Letter · Decision Letter 2]

18 Oct 2022

Contextual factors predicting compliance behavior during the COVID-19 pandemic: A machine learning analysis on survey data from 16 countries

PONE-D-21-27819R2

Dear Dr. Hajdu,

We’re pleased to inform you that your manuscript has been judged scientifically suitable for publication and will be formally accepted for publication once it meets all outstanding technical requirements.

Kind regards,

Tarik A. Rashid, PhD

Academic Editor

PLOS ONE

Reviewers' comments:

Reviewer's Responses to Questions

**Comments to the Author**

1. If the authors have adequately addressed your comments raised in a previous round of review and you feel that this manuscript is now acceptable for publication, you may indicate that here to bypass the “Comments to the Author” section, enter your conflict of interest statement in the “Confidential to Editor” section, and submit your "Accept" recommendation.

Reviewer #1: (No Response)

2. Is the manuscript technically sound, and do the data support the conclusions?

Reviewer #1: Yes

3. Has the statistical analysis been performed appropriately and rigorously? 

Reviewer #1: Yes

4. Have the authors made all data underlying the findings in their manuscript fully available?

Reviewer #1: No

5. Is the manuscript presented in an intelligible fashion and written in standard English?

Reviewer #1: Yes

6. Review Comments to the Author

Reviewer #1: Dear authors

The only requirement for this revision was high quality graphics. The graphics are still of very low quality. Please correct this. Taking screen shots is not professional - save the graphics appropriately with your software.

7. PLOS authors have the option to publish the peer review history of their article (what does this mean?). If published, this will include your full peer review and any attached files.

Reviewer #1: No

---

## [Editor Report · Acceptance letter]

15 Nov 2022

PONE-D-21-27819R2 

Contextual factors predicting compliance behavior during the Covid-19 pandemic: A machine learning analysis on survey data from 16 countries 

Dear Dr. Hajdu:

I'm pleased to inform you that your manuscript has been deemed suitable for publication in PLOS ONE. Congratulations! Your manuscript is now with our production department. 

Kind regards, 

on behalf of

Dr. Tarik A. Rashid 

Academic Editor

PLOS ONE